# Identification of Patients with Potential Atrial Fibrillation during Sinus Rhythm Using Isolated P Wave Characteristics from 12-Lead ECGs

**DOI:** 10.3390/jpm12101608

**Published:** 2022-09-29

**Authors:** Hui-Wen Yang, Cheng-Yi Hsiao, Yu-Qi Peng, Tse-Yu Lin, Lung-Wen Tsai, Chen Lin, Men-Tzung Lo, Chun-Ming Shih

**Affiliations:** 1Division of Sleep and Circadian Disorders, Departments of Medicine and Neurology, Brigham and Women’s Hospital, Boston, MA 02115, USA; 2Division of Sleep Medicine, Harvard Medical School, Boston, MA 02115, USA; 3Department of Biomedical Sciences and Engineering, National Central University, Taoyuan City 320317, Taiwan; 4Division of Cardiology, Department of Internal Medicine, Taipei Medical University Hospital, Taipei 110301, Taiwan; 5Cardiovascular Research Center, Taipei Medical University Hospital, Taipei 110301, Taiwan; 6Division of Cardiology, Department of Internal Medicine, School of Medicine, College of Medicine, Taipei Medical University, Taipei 110301, Taiwan; 7Taipei Heart Institute, Taipei Medical University, Taipei 110301, Taiwan; 8Department of Medicine Research, Taipei Medical University Hospital, Taipei 110301, Taiwan; 9Department of Information Technology Office, Taipei Medical University Hospital, Taipei 110301, Taiwan; 10Graduate Institute of Data Science, College of Management, Taipei Medical University, Taipei 110301, Taiwan; 11Department of Internal Medicine, School of Medicine, College of Medicine, Taipei Medical University, Taipei 110301, Taiwan

**Keywords:** atrial fibrillation, cardiovascular disease diagnosis, inter-lead dispersion, empirical mode decomposition, signal processing, machine learning, spike, impulse noise, physiological time series

## Abstract

Atrial fibrillation (AF) is largely underdiagnosed. Previous studies using deep neural networks with large datasets have shown that screening AF with a 12-lead electrocardiogram (ECG) during sinus rhythm (SR) is possible. However, the poor availability of these trained models and the small size of the retrievable datasets limit its reproducibility. This study proposes an approach to generate explainable features for detecting AF during SR with limited data. We collected 94,224 12-lead ECGs from 64,196 patients from Taipei Medical University Hospital. We selected ECGs during SR from 213 patients before AF diagnosis and randomly selected 247 age-matched participants without AF records as the controls. We developed a signal-processing technique, MA-UPEMD, to isolate P waves, and quantified the spatial and temporal features using principal component analysis and inter-lead relationships. By combining these features, the machine learning models yielded AUC of 0.64. We showed that, even with this limited dataset, the P wave, representing atrial electrical activity, is depicted by our proposed approach. The extracted features performed better than the bandpass filter-extracted P waves and deep neural network model. We provided a physiologically explainable and reproducible approach for classifying patients with AF during SR.

## 1. Introduction

Atrial fibrillation (AF) is the most common arrhythmia with an estimated prevalence of 1–4% in the general population [1]. Adults with AF have a 5-fold greater risk for stroke, 1.5-fold greater risk for all-cause mortality, increased risk of development and mortality from heart failure, and a higher risk for dementia [2,3,4]. Independent of other associated cardiovascular conditions, the quality of life of patients with AF is impaired [5]. The diagnosis of AF requires an electrocardiogram (ECG) to document the typical AF rhythm. For potential patients with AF symptoms, general procedures in hospitals usually involve an ECG Holter device that would be brought back home and used to record for >24 h [6,7]. The recent development of wearable devices with dry electrodes that enable patients to start recording only when symptoms occur [8,9] is another solution. However, some patients with AF go undiagnosed owing to the asymptomatic (‘silent AF’) and paroxysmal occurrence of AF [10]. Identifying and determining potential AF in patients during sinus rhythm (SR) in regular 12-lead ECG examinations is crucial and beneficial for further systematic screening.

There has been increasing evidence showing structural changes that occur in the atrium before AF development [11], and these structural changes may be reflected on the 12-lead ECG. Indeed, developing an artificial intelligence (AI) to either classify paroxysmal AF in SR or predict the risk of AF before its occurrence is reportedly possible [12,13,14,15]. However, these AI models are deep learning (DL)-based and trained on unpublished, large patient databases, and the trained model is not publicly accessible, limiting its reproducibility and clinical application. Additionally, the critical features contributing to AF risk are not clearly identified using the AI model. Even though a recent study [14] used salient mapping to highlight the most sensitive area in the ECG waveform that contributes the most to the prediction, the results highlighted the entire P wave (and part of the ST-segment), which does not provide further information regarding spatial and structural abnormalities from the informative 12-lead signal.

In this study, we propose a feature-based machine learning (ML) model that provides explainable results and can be trained on an achievable database size. The major challenge in identifying AF features is the non-prominent P waves on ECGs. Therefore, we developed a method to isolate P waves from ECG and formulated P wave-related features that are free from interference from QRS complexes. We then trained a ML model with these features using a database containing approximately 230 untreated patients with pre-diagnostic symptoms. We hypothesized that (1) without any drug interference, abnormalities occur in the 12-lead ECG signal within 90 days before AF diagnosis, (2) with a limited number of available data recordings, feature-based ML models exhibit better performance than DL models, and (3) in feature-based ML models, preprocessing of an ECG signal is a crucial step in deriving reliable features for better ML performance.

## 2. Materials and Methods

### 2.1. Subject Selection

We included all digitally available standard 10 s 12-lead ECG recordings from the Taipei Medical University Hospital between 29 January 2015 and 7 March 2020. A total of 94,224 digital ECG recordings from 64,196 patients were collected and de-identified. All ECGs were recorded in the supine position at a sampling rate of 500 Hz. Trained cardiologists labeled all recordings. The Taipei Medical University Hospital Review Board waived the requirement for informed consent.

To select the recordings that will be used to identify the characteristics of potential AF during SR, we identified patients who had at least one AF rhythm in these ECG recordings. Many patients underwent multiple recordings during the study period. To maintain the time dependence of the AF instance, we defined a collection window of 90-days prior to the first AF incident. We considered only ECG recordings marked as SR, and 287 recordings from 213 patients were selected according to these criteria. The last ECG recorded for each patient was included in the analysis (Figure 1).

Subjects in the control group were selected based on the following criteria: (1) each patient had at least two normal sinus ECG measurements; the interval between the first and second ECG was 2–6 months, (2) all ECG recordings from the same patient were SR, (3) patients had no ICD (International Classification of Diseases) 10 code of AF in their electronic medical record, and (4) subjects aged 50 years and older. A total of 6605 subjects met these criteria. To avoid the effect of age on ECG features, we used stratified sampling to ensure a similar age distribution between the AF and control groups. We set up age groups using a 10-year age bin. Within each age bin, we randomly sampled control subjects twice in the number of the AF group. This results in 580 participants in the control group. After the patient list was determined, the first ECG recording for each patient was chosen for analysis. However, we found that many of the ECGs contained strong artifacts that may affect the results and thus removed these ECGs. Finally, the total number of available ECGs was 247. The average age was 75.6 years (standard deviation (SD) = 12.7) in our AF group and 76.7 years (SD = 10.3) in our control group.

### 2.2. ECG Signal Processing

The scheme used in this study comprised signal processing, feature extraction, and statistical analysis. A complete flowchart of this study is shown in Figure 2. The details of each step are provided below. Signal processing and feature extraction were performed using MATLAB (v2021b) with previously developed scripts for MA-EMD (minimum-arclength uniform phase empirical mode decomposition) [16] and fast EMD [17].

#### 2.2.1. Noise Filtering

The ECGs were resampled at 500 Hz. First, we filtered the baseline wonder using discrete wavelet decomposition. The ECG was decomposed with a Symlet 10 wavelet at level 8. The eighth approximate coefficient was set to zero to remove the low-frequency component. The filtered signal was then reconstructed using inverse wavelet transform. Subsequently, a high-pass filter with a cut-off frequency of 32 Hz was applied. Some of the devices encountered artifacts in the first or last 0.5 s; these artifacts were removed. Therefore, the length of the ECG recordings available for analysis ranged from 6.6 to 10.9 s.

#### 2.2.2. ECG Delineation

To identify critical points on the ECG for later extraction of the P wave and other morphological ECG features, we used an open-source QRS detector and waveform limit locator, ECGPUWAVE [18], which has excellent performance for P wave and QRS detection. The Q-, R-, and S-waves and the onset and offset of the P waves were identified for further processing.

#### 2.2.3. P Wave Extraction Using Minimum-Arclength Uniform Phase Empirical Mode Decomposition (MA-UPEMD)

The P wave of an ECG reflects electrical activity originating from the atrium. Thus, in this study, we aimed to extract features from P waves. However, when applying feature extraction analysis, the results are easily influenced by the QRS and T waves, which dominate the ECG in amplitude and time, respectively. Therefore, we propose a novel algorithm to isolate P waves from the ECG such that the extracted feature is dependent only on the P wave. We introduced the MA-UPEMD algorithm, which can separate impulse-like noises (spikes) in a signal at a precise time and frequency range. Here, we propose a two-step algorithm. The first step involves removal of the QRS wave, which has a wide frequency spectrum and is spatially close to the P wave. In the second step, the P wave is further separated from the T wave using a similar technique.

MA-UPEMD is a modification of our previously developed MAMA-EMD (minimum-arclength masking EMD) [16] that replaces the masking method in EMD (empirical mode decomposition) [19] with uniform-phase method [20]. In brief, EMD is an adaptive, nonlinear, data-driven filter that decomposes a time series into several frequency-limited components called intrinsic mode functions, and in this algorithm, extrema are used to generate the mean of the upper/lower envelope depicting the low-frequency components in the signal; therefore, the extrema distribution determines the frequency response of the filter [21,22]. UP-EMD (uniform-phase EMD) improves spatial homogeneity of the frequency response by adding a set of sinusoidal signals (masking signals) of the same amplitude and frequency, but with phases uniformly distributed within 2π at each of the realizations; it has been shown to resolve mode splitting and residual noise [20]. Figure 3a shows an ECG signal, the sinusoid signal of one of the phases, and a summation of these two signals. However, an impulse-like spike (such as a QRS wave) has a wide frequency spectrum. The UP-EMD filters only the high-frequency component of the spike. Thus, we further introduce the minimum arclength criterion on the spike, which allows a spatial-specific filtering effect. With this minimum arclength criterion, a replacement is found for the extrema on the spike by minimizing the arclength of the envelope during the envelope depicting procedure (Figure 3b). In this manner, an impulse-like spike can be isolated from the signal.

We briefly describe the abovementioned steps in Figure 3. We took the averaged beat of each lead by aligning the beats with their R-peak positions (Figure 3a). Each patient then had a 1 s by 12-lead ECG wave matrix. MA-UPEMD was then performed on the averaged beat of each lead. The first MA-UPEMD was applied to remove the QRS wave from the ECG (Figure 3a,b); the sinusoids in UP-EMD were 30 Hz in frequency, the amplitude was 0.04, and the number of phases was four. The minimum arclength criterion was applied to the QRS points (Figure 3b). After performing MA-UPEMD, we derived a QRS-free ECG signal (Figure 3c) with P and T waves. The remaining P and T waves still encountered serious baseline drifts. Since we aimed to preserve the intact waveform of the P wave, we did not use Fourier-based linear filters and instead performed another MA-UPEMD. In this case, the P wave was the target for the minimum arclength criterion, and the frequency of the added sinusoid was 10 Hz, with an amplitude of 0.02 and four realizations. As a result, we isolated the P wave of the averaged beat for each of the 12 leads (Figure 3d,e).

#### 2.2.4. P Wave Extraction Using Bandpass Filter

To further demonstrate the superiority of our proposed method of P wave extraction, we used a conventional bandpass filter to extract P waves. After the same process of noise filtering, R-peak detection was implemented for beat alignment. The signal was filtered using a 4th-order Butterworth bandpass filter with a frequency range of 0.5–8 Hz, which was considered to be the frequency band of the P wave. After filtering, we aligned the beats using the original R-peak position and averaged all the beats in the same channel. The entire process of the bandpass filter method is shown in Figure 4. After P wave extraction using a bandpass filter, the same feature extraction was performed, and the ML models were trained separately on the features derived from the bandpass filter to compare the differences in performance.

### 2.3. Feature Extraction

#### 2.3.1. ECG Morphology Features

Conventional ECG morphological features were calculated from the ECG critical points detected using ECGPUWAVE [18]. Owing to collinearity, we chose the leads with the largest R amplitude among the limb and chest leads, and the morphological features were extracted on the averaged beat of the two chosen leads. The positions of the ECG critical points include the peak, onset, and offset of the P wave; onset and offset of the QRS complex; Q, R, and S waves; and the peak, onset, and offset of the T wave. The four types of ECG morphological features are defined below:Wave amplitude: the amplitudes of the P, Q, S, and T waves were defined by their peaks.P duration: we calculated the duration of the P wave as the time between onset and offset.Intervals: traditionally, ECG features are calculated and include the duration between P-onset and R-onset (PR interval), that between Q onset and J point (QRS duration), and that between Q onset and T offset (QT interval).ST-voltage: the height of the ECG segment between the J point and T onset, which is usually used for the diagnosis of myocardial infarction, was used. The average voltage between the two points was calculated (STvol_R).

#### 2.3.2. P Wave Projection by Principal Component Analysis (PCA)

To analyze the depolarization route of the atrium, we projected the 12-lead ECG onto a three-dimensional (3D) vector space spanned by the principal component (PC) of the ECG (Figure 5). This transformation can project the depolarization route better than traditional vectorcardiogram transforms because each participant has a unique P wave axis. This projection method has been used on the QRS complex to distinguish patients with arrhythmogenic right ventricular cardiomyopathy [23]. Here, we applied the same technique to the P wave-extracted ECG. We constructed the P wave matrix (*X*_nx8_) using eight of the 12 leads (I, II, V1, V2, V3, V4, V5, and V6) with *n* observations (*n* = 500 points, 1 s), and decomposed the correlation matrix (*X^T^X*) using PCA. The first three PCs—PC1, PC2, and PC3—account for 99% of the total variance and are representative of the route and variation of the P-loop. The weights of PC1, PC2, and PC3 (the eigenvalues) were included as a P wave feature from PCA (i.e., PC1w, PC2w, and PC3w).

#### 2.3.3. P-Loop Descriptors

Subsequently, the previously developed QRS-loop descriptors were applied to the P-loop. First, we took the Fourier transforms of PC1, PC2, and PC3, and calculated the ratio between the total power in these two frequency bands—20–50 Hz and 1–20 Hz (r2050_PC1, r2050_PC2, and r2050_PC3). Second, we calculated the area and length of the P-loop on the two-dimensional plane expanded by PC1 and PC2 (Figure 6). The minimum rectangle that encompasses the P-loop was divided into N cells (N = 4900 in this study) of equal size. The P-loop area (LoopArea) was defined as the percentage of cells inside the P-loop. This area represents the regularity of the loop and is reduced when convex and concave components exist in the loop. The P-loop length (LoopLength) was calculated as the total number of cells passing through the route. An increase in the P-loop length indicates the dispersion or inhomogeneity of the route. We also calculated the length-to-area ratio, which represents the roughness of the P trajectory.

#### 2.3.4. Inter-Lead P Wave Dispersion

We also measured the dissimilarity of the P-loop between patients with potential AF and normal individuals by analyzing inter-lead relationships. This was previously used in the QRS loop and was named the inter-lead QRS dispersion [23]. In this study, we applied the same concept to the P wave method. In the new 3D vector space of PCA, each lead was mapped to a vector in the new orthogonal axes constructed by the first three PCs. We calculated the angles between each pair of the reconstructed vectors (Figure 7). A smaller angle indicated spatially closer vectors and vice versa. The differences in angles between the AF and normal groups represent a change in the inter-lead relationship, which indicates a shape distortion in the P-loop.

### 2.4. Classification of Patients with AF

#### 2.4.1. ML Model in AF Prediction

To understand the predictability of these features, we performed classifications using ML models, including support-vector machine (SVM), random forest, perception, and extreme gradient boosting (XGBoost). We combined all 56 features (Table 1), including one P wave feature, (P wave duration); 18 morphological features (amplitudes of the P, Q, R, S, and T waves for the chest and limb leads; PR, QRS, QT, and ST intervals for the chest and limb leads); P loop area, length, and area/length ratio (three features); the weights of the first three PCs of the P loop (three features); inter-lead P wave dispersion for each pair of the 12 leads (28 features); and total power in the 20–50 Hz frequency band in the three PCs (three features). Furthermore, features with >90% of the data missing were dropped before the model training. Dropped features include the amplitude of Q and S waves for limb leads, amplitude of P, Q, and S waves, and PR interval for chest leads (six features). The final feature space contained 50 features. All missing values were imputed using the median of the features in the training data, and all features were z-score normalized.

Since P waves are considered the most important feature in clinical diagnosis for atrium-related diseases, we also performed ML classification using two features: P wave amplitude and duration for comparison.

Four types of ML classifiers were used to classify the AF: SVM, perceptron, random forest, and XGBoost. The SVM projects the samples to a high-dimensional space with a kernel function and maximizes the gaps between the groups in the space. Here, we used the radial basis function as the kernel function. A perceptron is the simplest artificial neural network with no hidden layers. Random forest and XGBoost are both ensemble learning methods based on decision trees. The random forest uses “bagging” methods (i.e., bootstrap subsamples of the whole population and a portion of the features) to generate multiple classification trees, and the final decision is the aggregation of all trees. XGBoost is the “boosting” approach of ensembled decision trees. Unlike random forest, which trains decision trees in parallel, gradient boosting trees build trees sequentially by placing more weight on the misclassified samples of the previous tree, and the final decision is the weighted summation of all the trees. All ML were performed using Python (version 3.6.9) with package scikit-learn 1.0.2 [24] for the SVM, perceptron, and random forest, and package xgboost (version 1.6.2) for XGBoost.

For model training and evaluation, the data were split into two sets: 80% of the data were in the training set, and 20% were in the holdout testing set. The parameters of the models were determined by grid search, using AUC (area under the receiver-operator curve) as a metric and training with 10-fold cross validation. To compare the different models, AUC, accuracy, sensitivity, specificity, precision, and F1 score were used to evaluate the performance of the selected models on the testing set.

#### 2.4.2. Feature Importance

One of the greatest advantages of feature-based ML is its explanatory ability. To identify the most important features in predicting AF, we derived the Shapley additive explanations (SHAP) [25] feature importance values using the open source python package SHAP. The program first calculated the Shapley values of the features of each instance in the dataset. Based on the coalitional game theory, the Shapley value of a feature (i.e., local feature importance) can be derived from the marginal contribution to the prediction (probability of AF, in our case) when adding the selected feature to the model with all other feature values randomly assigned. The SHAP value for global feature importance is then defined as the average absolute Shapley values per feature across the data. This SHAP value is different from the feature importance derived for tree-based classifiers (XGBoost or random forest) and can be calculated for all classifiers, including perceptron, SVM, XGBoost, and random forest.

#### 2.4.3. Comparison with DL

To compare the performance of feature-based ML and end-to-end DL, we constructed a convolutional neural network (CNN). Owing to the small dataset in our study, instead of constructing a complicated model, a relatively simple structure was constructed to prevent the model from overfitting (Appendix A). The input of the network is ECG signals with an input shape (5000, 12), meaning 5000 samples with 12 leads. The CNN model comprises two convolutional blocks, each of which is composed of a one-dimensional convolutional layer and a max-pooling layer with a pooling size of two. For the convolutional layers, the ReLU activation function was adopted with a filter length of five, starting with 16 filters in the first block and increasing to 32 filters in the second block. Following the two convolutional blocks for temporal feature extraction was a dropout layer with a dropout rate of 0.2. Then, a fully connected layer with 16 neurons was used to connect the spatial leads with the ReLU activation function and a dropout layer with a dropout rate of 0.5. These convolutional blocks and fully connected layers were used to combine the temporal and spatial information in the input 12-lead signals. We then used a dense layer to incorporate this information. Finally, the output layer of the network was a dense layer with a sigmoid function used to predict the probability of AF occurrence. The model was optimized using the Adam optimizer with a learning rate of 10^−5^ to minimize binary cross-entropy. The validation set was used to test the performance of the model after each epoch and for hyperparameter tuning. The CNN model was developed using Keras (version 2.4.0) with TensorFlow (version 2.4.1) backend in Python (version 3.6.9).

## 3. Results

### 3.1. Comparison of P Wave Signals Extracted by MA-UPEMD and Band-Pass Filter

To evaluate our proposed method for P wave extraction, the conventionally used bandpass-filtered method was also applied to the signal and compared in our study. The proposed method preserved more detailed fluctuations of the P wave than the bandpass-filtered signal (Figure 8a). In addition, the spectrograms of the P wave signal were derived using continuous wavelet transform to observe whether the P wave could be independently isolated in both the time and frequency domains (Figure 8b–d) by the two methods. The time-frequency representation of the R wave is a triangular mask across both the low- and high-frequency bands (Figure 8b). For such a wide-band waveform, the leakage of the R wave could not be removed using the conventional bandpass filter method. Furthermore, the P wave form was compromised and distorted by the leakage of the adjacent R wave owing to the undesired frequency response of the bandpass filter (Figure 8d). In contrast, the MA-UPEMD method largely prevented the leakage of the R wave and kept the P wave feature unharmed (Figure 8c).

### 3.2. Prediction of AF Using Different P Wave Extraction and ML Models

Table 2 shows the evaluation results of all selected models. All the scores were reported with a probability threshold selected using the Youden index on the training set (i.e., maximizing the sum of sensitivity and specificity). Among all the models, the perceptron achieved the highest AUC (0.64) (Table 2, Figure 9), with an F1-score of 0.61. The model had moderate specificity (0.71) but low sensitivity (0.47). Compared to the bandpass filter-derived P waves, whose highest F1 was 0.60 and AUC was 0.63, our proposed method improved the F1 score by 1%, AUC by 1%, and sensitivity by 7%.

Furthermore, the performance of the CNN, which was trained directly on the raw signal, was also not better than that of our proposed methods. The CNN model only gave an F1 score of 0.53 (Table 2), with an AUC of 0.51. Lower sensitivity (0.57) and specificity (0.52) were also obtained from the CNN model.

### 3.3. Classification with Only P Wave Amplitude and Duration

Table 3 and Figure 10 present the results for these ML classifiers with only the P wave amplitude and duration as inputs. All four classifiers gave an AUC < 0.56 and an F1-score < 0.57, both of which are lesser than those gotten when using all the features we derived.

### 3.4. Feature Importance

We further checked the SHAP feature importance values derived from the best model, perceptron, and the top ten features that contributed to the model are listed in Figure 11. The SHAP summary plot showing the relationship between the value of each feature and its impact on AF probability (positive/negative) is shown in Appendix A. In these models, the P wave angle between leads II and V4 (II_V4) was the most important, followed by the angle between leads V3 and V5 (V3_V5). The third important feature was the P wave duration (P_dur). Specifically, larger angles between leads II and V4; leads V3 and V5; and longer P wave duration contribute to a higher risk in AF (Appendix A).

## 4. Discussion

In this study, we developed an algorithm called MA-UPEMD to isolate P waves from 12-lead ECG recordings. With the features quantifying the temporal and spatial alterations of the P wave, the ML models achieved an AUC of 0.64. This shows that, first, without any drug effects, there are abnormalities in the 12-lead ECG for identifying potential AF in patients during SR, even among age-matched normal controls. Second, comparing the results of feature-based ML and DL, we found that ML performed better in this limited number of patients in the database and provided explainable results for clinical inferences. Third, the feature extraction step in ML further highlighted the importance of signal preprocessing. Comparing the results of MA-UPEMD and the bandpass filter, the proposed MA-UPEMD better depicted the P waveforms and resulted in higher classification scores.

Our perceptron model gives an AUC of 0.64, a specificity of 0.47, a sensitivity of 0.71, and a precision of 0.54, maximizing the sum of sensitivity and specificity, that is, the Youden index. Note that this cutoff can be adaptively changed according to the application scenario. For example, one can have a cutoff for higher sensitivity but lower precision as a pre-screening step to identify more potential patients. Then, a 24 h Holter or wrist-worn ECG event recorder can be provided for the patient. A recently published example is the optical-based irregular pulse notification algorithm of a smart watch proposed by Apple [26]. The algorithm provided a low precision of 0.34, thus achieving a high sensitivity of 0.84. Considering the high risks and burden of stroke in patients with AF [27,28], early identification of potential patients can be performed with a low-cost 12-lead ECG device for further anticoagulation treatment.

We also showed that with limited samples in the database, a feature-based ML model performed better than the DL model. Thus, preprocessing the signal and deriving physiologically meaningful features are crucial to the final performance of classifiers. We proposed several features (adapted from the previously published QRS inter-lead analysis [23]) to quantify the abnormality of the atrial activity from this extracted P wave, including the weights of each PC, the angles between each pair of leads when projecting onto the PCs, and the loop length and distance of the P wave. These features contributed significantly to the ML models and increased the classification AUC compared with those using only P wave duration and amplitude (AUC = 0.56 using P wave duration and amplitude; AUC = 0.64 using all features). Specifically, the angles between leads II and V4 and leads V3 and V5 are the top two important features (a larger value indicating higher AF risk). The larger angles in these two pairs of leads are indications of distortion of the electrical conduction, which may be due to abnormal atrial structure or triggers [29]. In addition, the P wave duration ranked the third important feature in our model, with a larger value indicating higher AF risk. The prolonged P wave duration, which results from enlarged atria or atrial heterogeneity, is known to be associated with AF [30]. However, the P wave duration alone does not provide sufficient predictability in our results. This highlights the significance of our proposed features.

Given the moderate AUC and accuracy of our ML on this small dataset, our findings showed limited power for the proposed features in detecting patients with AF during sinus rhythm. Since the pathology of AF involves fibrosis of any part of the atria, a single source of abnormality or single-lead distortion may be under-detected by our features which perform PCA on all leads and analyze inter-lead relationships. Several studies using deep convolutional neural network models on each lead have detected subtle changes in the early stages of AF. These deep convolutional neural network models (DNN) were trained with large databases containing >650,000 ECG recordings, providing a holistic and comprehensive view of the pathology of AF and therefore reaching an AUC > 0.8 [12,13,14]. Note that the DNN model performance varies according to sex, age, and ethnic groups [12] and decreases when transferring databases [14]. Building a model with high performance for clinical use requires expanding the database on the target population as well as a data-efficient architecture/approach reflecting the pathology of the target disease to maximize the available data. Considering the difficulties in data collection in biomedical science, our explainable ML model, which underlies the key features in AF detection, may also provide a reference for constructing models with higher efficiency and accuracy.

## 5. Limitations

Our study has some limitations. First, the ECGs were collected from patients visiting Taipei Medical University Hospital for clinics or inpatient care; the prevalence of AF (3.5%) in our study population was higher than the prevalence of approximately 1.07% in the general population [31]. Second, although we only selected ECG records obtained within three months prior to AF diagnosis, patients may have had various unknown lengths of AF, which further increases the variability in our data. Third, we chose age-matched patients as controls so that the trained models depicted abnormalities resulting merely from AF and not from aging. This restricts the applicability of our results to general populations with younger ages. Including younger patients might increase the AUC/accuracy and give a seemingly better model performance due to high specificity in the young population. Indeed, a preliminary result including younger patients in the control group from the same cohort increased the AUC to approximately 0.8 (unpublished data). However, our study focused on a population with a higher risk for AF due to more explainable differences in features and realistic scenarios. Furthermore, given the prevalence of AF in the population aged >65 years, some patients in our control group may have also had paroxysmal AF, which was not detected in all visits. Finally, the higher chance of AF occurrence predicted by the ML model does not justify anticoagulation or antiarrhythmic therapy, at least until now. Once a high-risk patient is identified, AF can be confirmed by long-term monitoring using conventional wearable devices [9,32], and the risk of stroke can be further evaluated by CHA_2_DS_2_-VASc Score or AF burden.

## 6. Conclusions

In conclusion, our study showed that with appropriate preprocessing and feature extraction, constructing a model on a limited database to identify potential AF in patients is possible. The proposed methods in this study provide physiologically explainable differences in the disease, work on a database of sizes that can be reproduced by regional hospitals and may serve as a reference for more sophisticated and efficient classification models. However, the accuracy of our proposed model is limited. Clinical diagnosis of AF should be performed with long-term monitoring before treatment. Nevertheless, our findings provide critical proof-of-concept evidence for unique information that might aid in identifying paroxysmal AF and should be further tested in different age groups, centers, and populations.

## Figures and Tables

**Figure 1 jpm-12-01608-f001:**
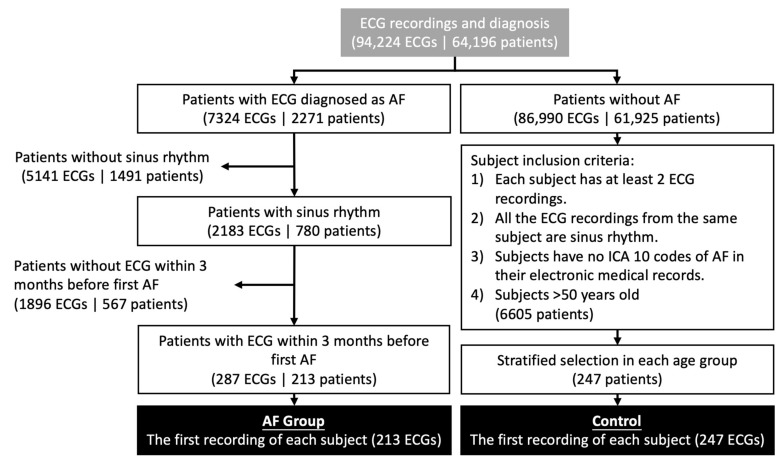
Patient selection for the AF and control groups.

**Figure 2 jpm-12-01608-f002:**
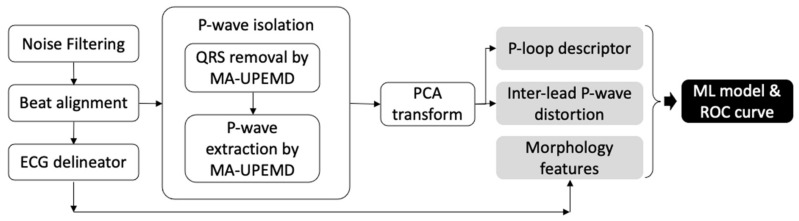
Flowchart of ECG processing and feature extraction in this study.

**Figure 3 jpm-12-01608-f003:**
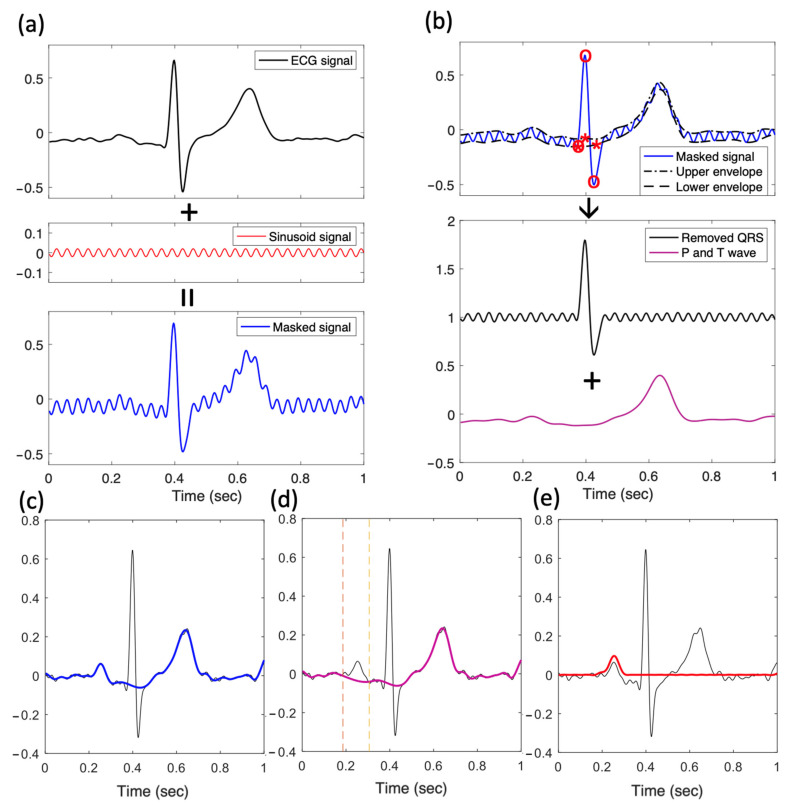
P wave isolation by applying the MA-UPEMD method. (**a**) The first step in MA-UPEMD: a sinusoid signal is added on the input signal, resulting in a masked signal. (**b**) Plots demonstrating one of the sifting iterations in MA-UPEMD. UPEMD is applied with the minimum arclength criterion on the QRS peaks. Specifically, when interpolating the spline for upper and lower envelope, the knots for Q, R, and S peaks (o) are replaced with new knots found by applying the minimum arclength criterion (*). After taking the mean of the upper and lower envelopes, the P and T waves are derived (purple). Subtracting this mean gives the QRS wave (black). This is only one sifting for one phase in MA-UPEMD. The process is repeated four times with different uniform phases to derive the mean signal with only P and T waves (**c**). Then, the same MA-UPEMD is applied again on the signal with P and T waves, with P wave as the spike to be separated. This results in the delineated baseline wonder and T wave (purple) in (**d**) and the extracted P wave (red) in (**e**).

**Figure 4 jpm-12-01608-f004:**
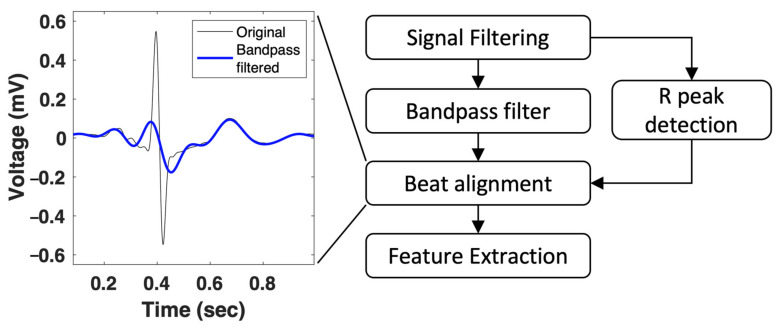
P wave extraction using the bandpass filter method.

**Figure 5 jpm-12-01608-f005:**
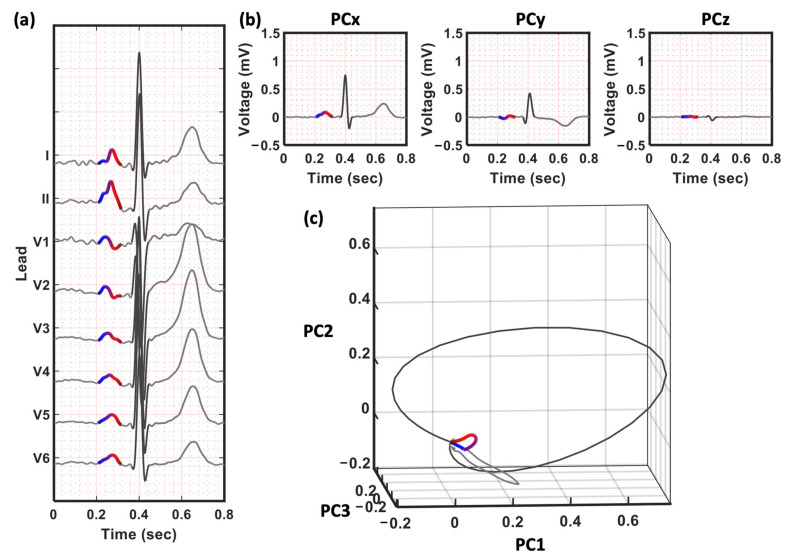
The PCA transform for eight of the 12 leads. (**a**) The eight channels for PCA transform (**b**) The 3 PCs—PC1, PC2, and PC2, respectively. (**c**) The three-dimensional space constructed by the new eigen space. The trajectory is defined by the three PCs in (**b**). Note that only the P wave is used when calculating the transformation matrix in PCA. The QRS and T waves in (**b**,**c**) are projected by the same matrix constructed by P wave and are only for demonstrating the relative position of the P wave.

**Figure 6 jpm-12-01608-f006:**
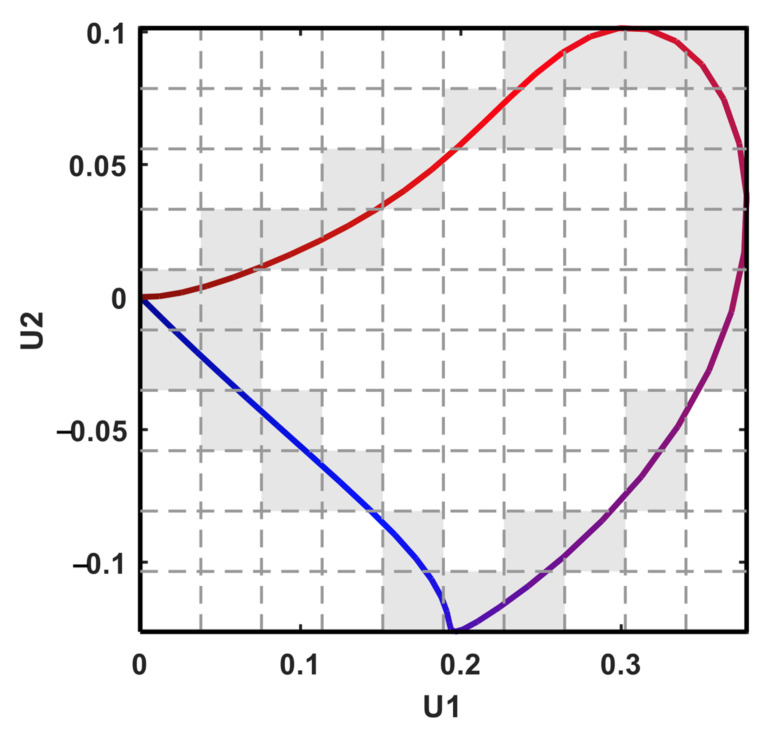
The P-loop descriptor by PC1 and PC2. The number of grey cells is the loop length, while the number of white cells inside the loop is the loop area.

**Figure 7 jpm-12-01608-f007:**
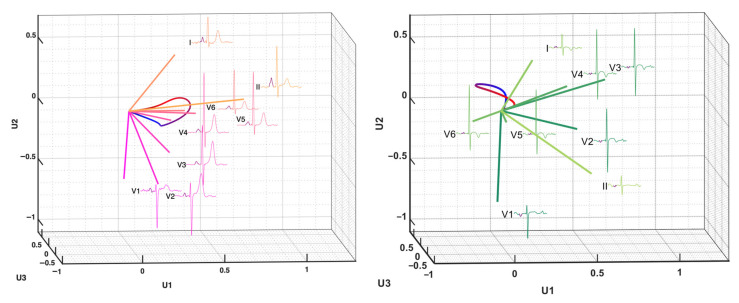
The inter-lead correlation of the P-loop by PC1, PC2 and PC3. The left panel is an example from a control subject and the right panel is an example from a patient in the AF group.

**Figure 8 jpm-12-01608-f008:**
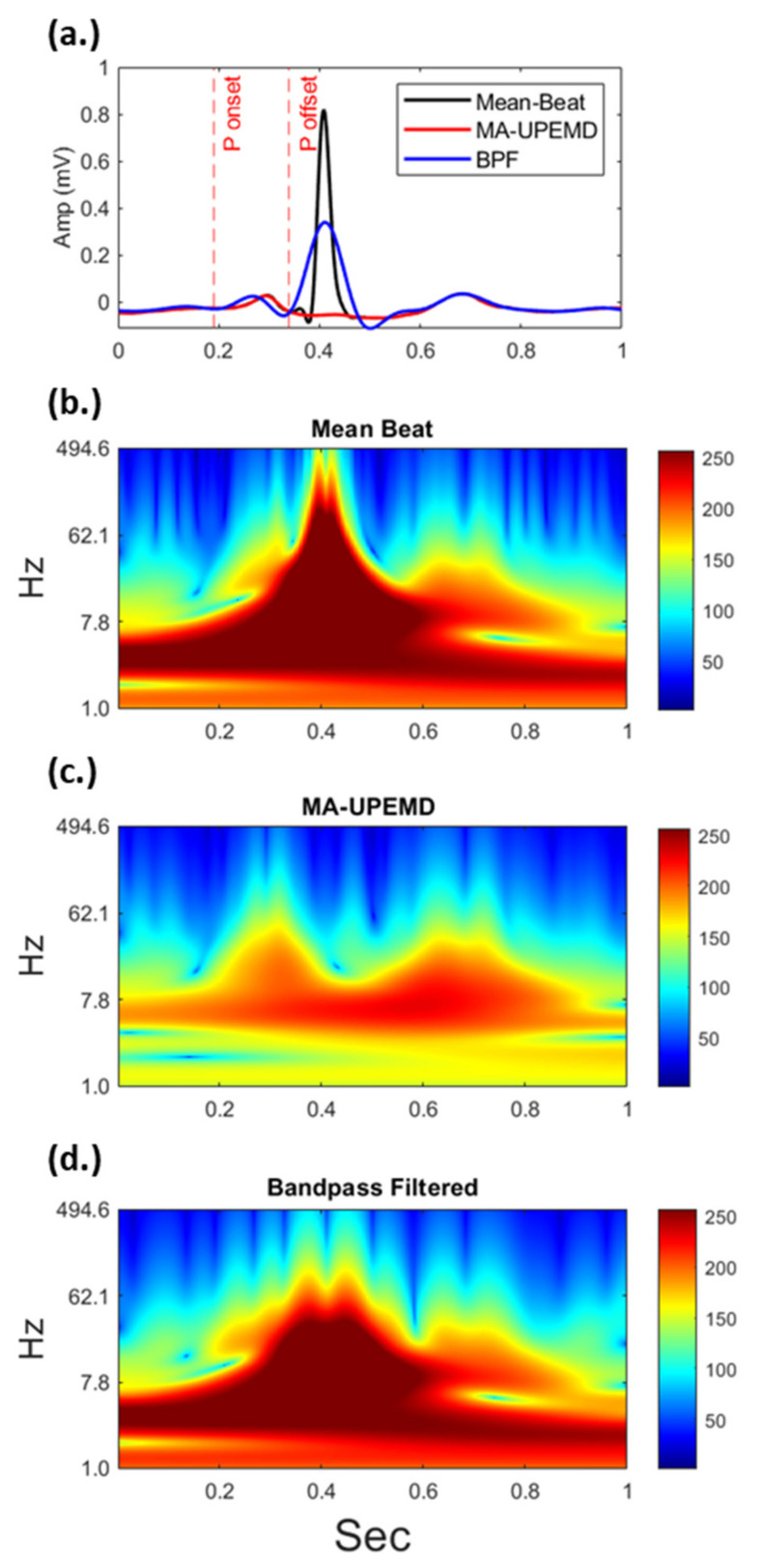
Comparison of P wave signals of the same ECG extracted by MA-UPEMD and the bandpass filter method in the time domain (**a**) and their corresponding time-frequency representations (**b**–**d**). The adjacent R wave, which has a wide frequency band as shown in (**b**), introduces power leakage in the bandpass filter method (**d**). In contrast, MA-UPEMD separates the R wave at a precise time and frequency range, and thus better preserves the P wave.

**Figure 9 jpm-12-01608-f009:**
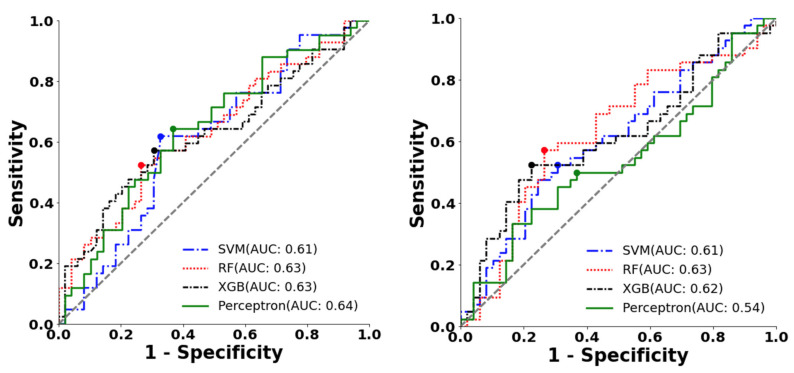
ROC curve and the optimal operating point of machine learning classifiers for AF prediction on the testing dataset. This figure shows the ROCAUC of SVM, random forest (RF), XGBoost (XGB), and perceptron classifiers using (**a**). features extracted with the proposed method (**b**). features extracted with the bandpass filter method.

**Figure 10 jpm-12-01608-f010:**
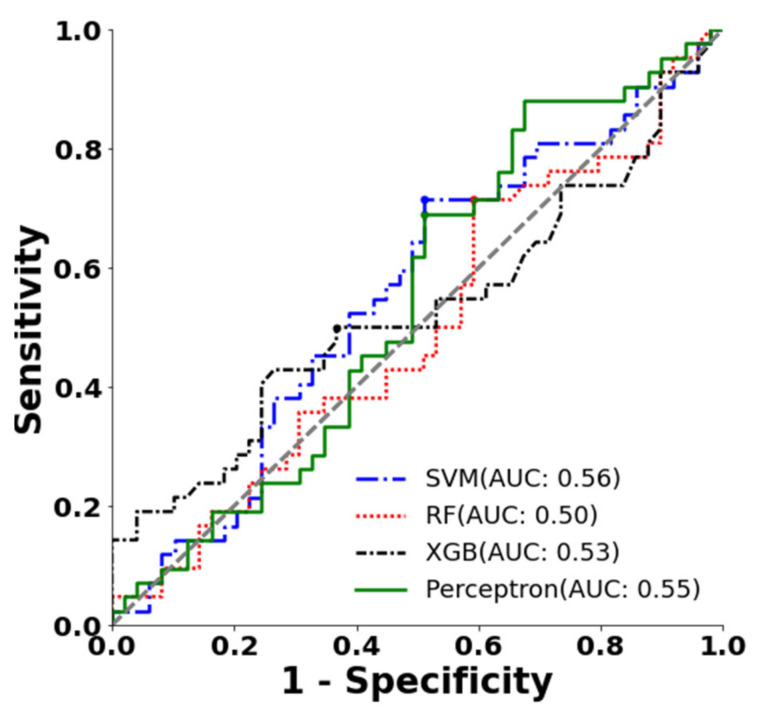
ROC curve and the optimal operating point of different machine learning classifiers (SVM, random forest (RF), XGBoost (XGB)) with P wave amplitude and duration.

**Figure 11 jpm-12-01608-f011:**
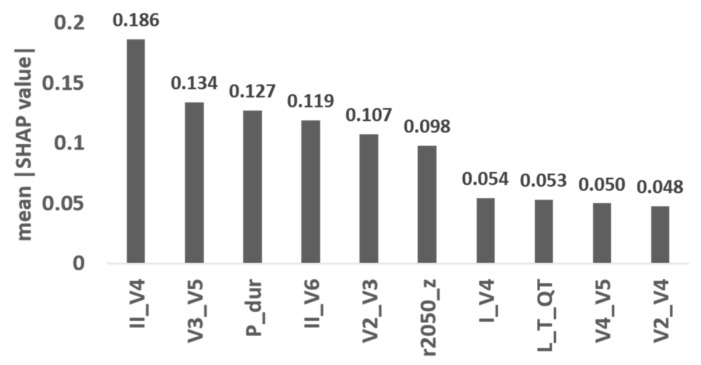
The top ten features that contribute the most SHAP feature importance in perceptron model. The abbreviations of feature names are explained in Table 1.

**Table 1 jpm-12-01608-t001:** List of features used in the machine learning models.

Feature	Feature Name	Description
P Wave	P_dur	P wave duration.
Morphology	L_AMP_X, C_AMP_X X may be P, Q, R, S, or T	Mean of P, Q, R, S, or T wave amplitude on limb leads and chest leads.
	L_T_X, C_T_X X: PR, QRS or QT	Mean of PR, QRS, or QT interval on limb leads and chest leads.
	L_STvol_R, C_STvol_R	Average voltage of ST segment on limb leads and chest leads.
Principal component	PC1w, PC2w, PC3w	Weight of eigenvalues from PC1, PC2 and PC3.
Inter-lead P wave dispersion	I_II, II_V1, V1_V2, V2_V3, V3_V4, V4_V5, V5_V6, I_V1, II_V2, V1_V3, V2_V4, V3_V5, V4_V6, I_V2, II_V3, V1_V4, V2_V5, V3_V6, I_V3, II_V4, V1_V5, V2_V6, I_V4, II_V5, V1_V6, I_V5, II_V6, I_V6	P-loop angle between two leads.
Loop analysis	LoopArea	Area of P-loop.
	LoopLength	Length of P-loop
	LAratio	Ratio of P-loop length and area
Frequency analysis	r2050_PC1, r2050_PC2, r2050_PC3	The ratio between the total power in frequency bands 20–50 Hz and 1–20 Hz

**Table 2 jpm-12-01608-t002:** Performance of the ML models on the holdout testing set.

Model	Method	AUC	Accuracy	Sensitivity	Specificity	Precision	F1
SVM	MA-UPEMD	0.61	0.58	0.67	0.51	0.55	0.60
	BP-filter	0.61	0.55	0.62	0.49	0.51	0.56
Random forest	MA-UPEMD	0.63	0.59	0.57	0.61	0.56	0.56
	BP-filter	0.63	0.60	0.64	0.57	0.56	0.60
XGBoost	MA-UPEMD	0.63	0.62	0.48	0.74	0.61	0.53
	BP-filter	0.62	0.50	0.50	0.50	0.46	0.48
Perceptron	MA-UPEMD	0.64	0.58	0.71	0.47	0.54	0.61
	BP-filter	0.54	0.51	0.50	0.51	0.47	0.48
CNN		0.51	0.54	0.57	0.52	0.50	0.53

**Table 3 jpm-12-01608-t003:** Performance of the ML models on the holdout testing set with only P wave amplitude and duration.

Model	AUC	Accuracy	Sensitivity	Specificity	Precision	F1
SVM	0.56	0.56	0.52	0.59	0.52	0.52
Random forest	0.50	0.48	0.43	0.53	0.44	0.43
XGBoost	0.53	0.59	0.43	0.74	0.58	0.49
Perceptron	0.55	0.56	0.62	0.51	0.52	0.57

## Data Availability

The data that support the findings of this study are available from the corresponding author upon reasonable requests.

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
