# Peer review of "Identification of Patients with Potential Atrial Fibrillation during Sinus Rhythm Using Isolated P Wave Characteristics from 12-Lead ECGs"

_jpm, 2022, doi:10.3390/jpm12101608_

Round 1

Reviewer 1 Report

The article proposed a method to identify AF from ECG recording. However the proposed method showed poor performance. The accuracy and sensitivity were less than 0.70 and 0.56, respectively. This performance was far from an acceptable performance. This performance was also far less than the methods presented in published articles.    

Author Response

We thank the reviewer's comment. However, unlike many articles trying to classify AF and non-AF rhythms from an ECG reading, this study aims to predict/identify subjects with paroxysmal AF during sinus rhythm. Indeed, recent studies using convolutional neuron networks with huge longitudinal databases have shown this is possible. However, these databases were hard to derive, and the model parameters were not shared. This limited the reproducibility of the results on other databases. Here we proposed a feature-based approach that provides explainable results reproducible on the scale that most local hospitals can achieve. We agree with the reviewer that the accuracy and sensitivity can be much improved by expanding the size of the database to include more variations in the heterogeneity in paroxysmal AF. This limitation is included in the last paragraph of the discussion.

“Given the moderate AUC and accuracy of our ML on this small dataset, our findings showed limited power for the proposed features in detecting patients with AF during sinus rhythm. Because the pathology of AF involves fibrosis of any part of the atria, a single source of abnormality or single-lead distortion may be under-detected by our features which perform PCA on all leads and analyze inter-lead relationships.”

Reviewer 2 Report

Dear editors,  

It is my pleasure to review the manuscript entitled “Identification of Potential AF Patient during sinus rhythm with isolated P-wave characteristics from 12-lead ECG”. The authors report a model to predict presence of AF based on p-waves observed during sinus rhythm. The paper has important value and prediction of AF with ECG during sinus rhythm is intriguing since large portion of AF is paroxysmal in nature. However, the paper has major limitations.  

#1. ROC curve is presented for machine learning classifiers such as SVM, RF, SGB, and Perception. However, it is not what clinicians want. We want to know which ECG parameters can predict the presence of AF. Such as p wave duration, amplitude, morphology…etc.  

#2. This paper mainly of machine learning process. I do not expertise in such area. Another reviewer who expertise in artificial intelligence is needed.  

#3. The clinical significance of “AI predicted AF” needs to be studied in future trials. Higher chance for AF occurrence does not justify anticoagulation or antiarrhythmic therapy, at least until now. This issue should be discussed in the limitation section.  

#4. Why did the authors include the age parameter in machine learning model? Age is the single most potent risk factor for AF. But, it is a clinical parameter and not ECG parameter. We want to know the capability of AI to predict chance of AF presence based on ECG and not by age.  

#5. Why did the authors only select 247 patients among 61,925 patients in non-AF group? What was the criteria for selection?  

#6. Most clinicians are used to observe p wave amplitude, morphology, and duration. Can authors provide AUC curve for such friendly and readily usable parameters?  

#7. I think abstract is too long and difficult to understand. Please make sure this paper have to be understandable by physicians and especially cardiologists.

Author Response

Reviewer 2

#1. ROC curve is presented for machine learning classifiers such as SVM, RF, SGB, and Perception. However, it is not what clinicians want. We want to know which ECG parameters can predict the presence of AF. Such as p wave duration, amplitude, morphology…etc.

Ans:

We thank the reviewer for this valuable suggestion. After deriving the model, we attempt to understand the contribution of each input variable in the prediction by SHAP index, which represents the contribution of each variable in prediction AF in the machine learning model. We have provided the top 10 features contributing to the model in the results (Figure 10). It shows that P duration ranked 3rd in the important feature. Specifically, a longer P duration contributed to a larger risk for AF (Supplementary Figure S2). As per the reviewer's suggestion, we added these results in section 3.4 and discussion.

Section 3.4 of the results:

“We further checked the SHAP feature importance values derived from the best model, perceptron, and the top 10 features that contributed to the model are listed in Figure 10. The SHAP summary plot showing the relationship between the value of each feature and its impact on AF probability (positive/negative) is shown in Supplementary Figure 2. In these models, the P wave angle between leads II and V4 (II_V4) was the most important, followed by the angle between leads V3 and V5 (V3_V5). The third important feature was the P wave duration (P_dur). Specifically, larger angles between leads II and V4, leads V3 and V5, and longer P wave duration contribute to a higher risk in AF (Figure S2).”

Paragraph 3 in discussion:

“… Specifically, the angles between leads II and V4 and leads V3 and V5 are the top two important features (a larger value indicating higher AF risk). The larger angles in these two pairs of leads are indications of distortion of the electrical conduction, which may be due to abnormal atrial structure or triggers [29]. In addition, the P wave duration ranked the third important feature in our model, with a larger value indicating higher AF risk. The prolonged P wave duration, which results from enlarged atria or atrial heterogeneity, is known to be associated with AF [30]. However, the P wave duration alone does not provide sufficient predictability in our results. This highlights the significance of our proposed features.”

#2. This paper mainly of machine learning process. I do not expertise in such area. Another reviewer who expertise in artificial intelligence is needed.

Ans:

We appreciate the reviewer’s valuable comments from a clinician’s view.

#3. The clinical significance of “AI predicted AF” needs to be studied in future trials. Higher chance for AF occurrence does not justify anticoagulation or antiarrhythmic therapy, at least until now. This issue should be discussed in the limitation section.

Ans:

We thank the reviewer’s comment and have added this concern in the limitation:

“…Finally, the higher chance of AF occurrence predicted by the ML model does not justify anticoagulation or antiarrhythmic therapy, at least until now. Once a high-risk patient is identified, AF can be confirmed by long-term monitoring using conventional wearable devices [9,32], and the risk of stroke can be further evaluated by CHA₂DS₂-VASc Score or AF burden.”

#4. Why did the authors include the age parameter in machine learning model? Age is the single most potent risk factor for AF. But, it is a clinical parameter and not ECG parameter. We want to know the capability of AI to predict chance of AF presence based on ECG and not by age.

Ans:

We thank the reviewer's comment. We removed age from the machine learning models and found that the model accuracy still holds. We derived an AUC = 0.64 in the perceptron model without age (as compared to 0.65 when including age). Indeed, in our previous results, age was not the most significant feature; it ranked > 10 in XGBoost and No. 9 in the SVM model. Accordingly, we have modified the related results in the manuscript, including the AUC, accuracy, sensitivity, specificity, and F1 in Table 2, the AUC curve in Figure 9, the feature importance ranking in Figure 10, and the supplementary materials.

#5. Why did the authors only select 247 patients among 61,925 patients in non-AF group? What was the criteria for selection?

Ans:

Among the 61925 patients, we first generate a list of patients for the control group based on the following criteria: (1) each patient had at least two normal sinus ECG measurements separating two to six months, (2) all ECG recordings from the same patient were SR, (3) patients had no ICD (International Classification of Diseases) 10 code of AF in their electronic medical record, and (4) subjects aged 50 years and older. A total of 6605 subjects met these criteria. To avoid the effect of age on ECG features, we used stratified sampling to ensure a similar age distribution between the AF and control groups. We set up age groups using a 10-year age bin. Within each age bin, we randomly sampled the number of control subjects at two times that of patients in the AF group. This results in 580 participants in the control group. After the patient list was determined, the first ECG recording for each patient was chosen for analysis. However, we found that many of the ECGs contained strong artifacts that may affect the results and thus removed these ECGs. Finally, the total number of available ECGs was 247. The average age was 75.6 years (standard deviation (SD) = 12.7) in our AF group and 76.7 years (SD = 10.3) in our control group.

AF group

Control group

AF group

Control group

Age Group

Number of subjects meets criteria

Number of subjects meets criteria

Number of subjects selected

Number of subjects passes QC

Number of subjects passes ECG quality control

20-20

2

0

0

1

 0

30-39

2

0

0

1

 0

40-49

7

0

0

3

 0

50-59

23

2080

46

19

16

60-69

53

2176

106

37

42

70-79

88

1430

176

61

76

80-89

91

740

182

65

92

90-99

33

167

66

26

19

>100

2

12

4

 0

2

total number of subjects

299

6605

580

213

247

#6. Most clinicians are used to observe p wave amplitude, morphology, and duration. Can authors provide AUC curve for such friendly and readily usable parameters?

Ans:

We thank the reviewer's suggestion. In the modified manuscript, we added three models to investigate the AUC with only P wave features: P amplitude alone, P duration alone, and P amplitude and P duration together. It showed that AUCs with only P wave amplitude and/or duration are generally lower than with all the proposed features (figure below): AUC with P-wave duration is 0.51; AUC with only P wave amplitude is 0.44; AUCs with both P wave duration and amplitude are less than 0.56 in all the models (table below), which is smaller than that derived with all the proposed features. We added the classification with both P-amplitude and duration in our manuscript as a comparison (Section 3.4) and highlighted the contribution of our proposed features in the discussion (paragraph 3).

Figure: (a) AUC curves for classifying AF with P wave amplitude and P wave duration, respectively. (b) AUC curves for ML classifiers with P-wave amplitude and duration as inputs.

Table 3. Performance of the ML models on the holdout testing set with only P wave amplitude and duration.

Model

AUC

Accuracy

Sensitivity

Specificity

Precision

F1

SVM

0.56

0.56

0.52

0.59

0.52

0.52

Random Forest

0.50

0.48

0.43

0.53

0.44

0.43

XGBoost

0.53

0.59

0.43

0.74

0.58

0.49

Perceptron

0.55

0.56

0.62

0.51

0.52

0.57

#7. I think abstract is too long and difficult to understand. Please make sure this paper have to be understandable by physicians and especially cardiologists.

We thank the reviewer’s suggestion and have modified the abstract.

Reviewer 3 Report

2. Materials and Methods

The authors should have specified in the final part of the introduction, where they specify how the proposed study is to be carried out, that the implementation takes place in the Python programming language. They only speculate a little about Python in subchapters 2.4.2. Feature importance and 2.4.3. Comparison with DL. 

3.2. Prediction of AF using different P-wave extraction and different ML models

AUC is a measure of the performance of a classifier. In the case of an ideal classifier, AUC = 1 and AUC = 0.5 indicates a classifier that randomly assigns observations to classes. Therefore, given that this study obtained a value equal to 0.65 for the AUC for the SVM model when using the MA-UPEMD method, we consider this to be in a good area, although this model should be further improved. Also, Table 2 shows the values obtained for the Perceptron model, although the authors did not specify anything about it in the previous chapters. 

5. Conclusions

Some of the details presented in Chapter 4. Discussions should have been punctuated in the final chapter of conclusions and here we refer to the limitations of the proposed models, but also to the future directions of the present study.

Author Response

Reviewer 3

  1. Materials and Methods

The authors should have specified in the final part of the introduction, where they specify how the proposed study is to be carried out, that the implementation takes place in the Python programming language. They only speculate a little about Python in subchapters 2.4.2. Feature importance and 2.4.3. Comparison with DL.

We thank the reviewer for addressing this issue. The signal processing and feature extraction is performed on MATLAB; the machine learning classifiers were performed with packages Scikit-learn and XGBoost on Python; the SHAP feature importance was using package SHAP on Python. We have added the description for which software and packages were used when they were first mentioned in the manuscript.

3.2. Prediction of AF using different P-wave extraction and different ML models

AUC is a measure of the performance of a classifier. In the case of an ideal classifier, AUC = 1 and AUC = 0.5 indicates a classifier that randomly assigns observations to classes. Therefore, given that this study obtained a value equal to 0.65 for the AUC for the SVM model when using the MA-UPEMD method, we consider this to be in a good area, although this model should be further improved. Also, Table 2 shows the values obtained for the Perceptron model, although the authors did not specify anything about it in the previous chapters.

We thank the reviewer's comments on considering this to be in a good area. A perception model is a particular case of an artificial neuron network (ANN) with no hidden layers. While this model is described in the section "2.4.1 Machine learning model in AF prediction" in our previous version, we agree that we should elaborate on more details and be consistent with the names. We have edited the methods. We apologize for this confusion.

  1. Conclusions

Some of the details presented in Chapter 4. Discussions should have been punctuated in the final chapter of conclusions and here we refer to the limitations of the proposed models, but also to the future directions of the present study.

We thank the reviewer’s suggestion and have added the limitations and future directions in the conclusion

…However, the accuracy of our current model is limited. Clinical diagnosis of AF should be performed with long-term monitoring before any treatments. Nevertheless, our findings provide critical proof-of-concept evidence for unique information that might aid in identifying paroxysmal AF and should be further tested on different age groups, centers, and populations.”

Round 2

Reviewer 2 Report

The authors improved their manuscript. I recommend acceptance. Thank you.